# Efficacy of Chenodeoxycholic Acid and Ursodeoxycholic Acid Treatments for Refractory Functional Dyspepsia

**DOI:** 10.3390/jcm11113190

**Published:** 2022-06-02

**Authors:** Sung Ill Jang, Tae Hoon Lee, Seok Jeong, Chang-Il Kwon, Dong Hee Koh, Yoon Jae Kim, Hye Sun Lee, Min-Young Do, Jae Hee Cho, Dong Ki Lee

**Affiliations:** 1Department of Internal Medicine, Gangnam Severance Hospital, Yonsei University College of Medicine, Seoul 06273, Korea; aerojsi88@gmail.com (S.I.J.); dmy24@yuhs.ac (M.-Y.D.); dklee@yuhs.ac (D.K.L.); 2Department of Internal Medicine, Soonchunhyang University College of Medicine, Cheonan Hospital, Cheonan 31151, Korea; taewoolee9@gmail.com; 3Department of Internal Medicine, Inha University School of Medicine, Incheon 22332, Korea; inos@inha.ac.kr; 4Digestive Disease Center, CHA Bundang Medical Center, CHA University, Seongnam 13496, Korea; endoscopy@cha.ac.kr; 5Department of Internal Medicine, Hallym University Dongtan Sacred Heart Hospital, Hallym University College of Medicine, Hwaseong 18450, Korea; dhkoh@hallym.or.kr; 6Department of Internal Medicine, Gachon University College of Medicine, Gil Medical Center, Incheon 21565, Korea; yoonmed@gachon.ac.kr; 7Biostatistics Collaboration Unit, Yonsei University College of Medicine, Seoul 06273, Korea; hslee1@yuhs.ac

**Keywords:** functional dyspepsia, biliary dyspepsia, gallbladder dyskinesia, litholytic agent

## Abstract

Refractory functional dyspepsia (RFD) is diagnosed when symptoms persist for at least 6 months despite at least two medical treatments. No consensus treatment guidelines exist. The implicated causes of functional biliary dyspepsia are a narrowed cystic duct, Sphincter of Oddi dysfunction, microlithiasis, and gallbladder dyskinesia. We investigated the treatment effects of litholytic agents. RFD patients were prospectively enrolled in six tertiary medical centers. All subjects took chenodeoxycholic and ursodeoxycholic acids (CNU) twice daily for 12 weeks. We monitored their medication adherence, laboratory results, and complications. The 7-point global symptom scale test scores were determined before and after treatment. Of the 52 patients who were prospectively screened, 37 were included in the final analysis. The mean age was 51.3 years: 14 were males, and 23 were females. Before treatment, the mean number and duration of symptoms were 2.4 and 48.2 months, and a mean of 3.3 FD-related drugs were taken. The mean CNU adherence was 95.3%. The mean global symptom scale score decreased from 5.6 pretreatment to 2.6 posttreatment. The symptom improvement rate was 94.6% (35 out of 37 patients). The only adverse event was mild diarrhea (10.8%) that was resolved after conservative management. Conclusions: CNU improved the symptoms of RFD patients who did not respond to conventional medications. Litholytic agents are good treatment options for patients with RFD and biliary dyspepsia secondary to biliary microlithiasis. Further prospective, large-scale mechanistic studies are warranted.

## 1. Introduction

The prevalence of dyspepsia in general populations ranges from 20% to 40% [1]. Functional dyspepsia (FD) refers to dyspepsia of no known cause, despite the performance of standard diagnostic tests [2]. FD is the most common type of dyspepsia (8–12% of populations) [3], characterized by epigastric pain or discomfort originating from the stomach or duodenum in the absence of any causative organic or metabolic disease [2]. The “Rome IV diagnostic criteria” for FD require the presence of one or more symptoms (including bothersome postprandial fullness, early satiety, epigastric pain, and epigastric soreness) that commenced at least 6 months before diagnosis and continued for the past 3 months [4]. The pathogenesis of FD is not fully understood. Standard therapies include prokinetics, analgesics, histamine 2 (H2)-receptor antagonists, proton pump inhibitors, antacids, serotonin-receptor antagonists, and antidepressants. Despite (supposedly) appropriate treatment, some patients continue to have symptoms; the condition is termed refractory FD (RFD). The current therapies work poorly in such patients and may trigger drug-related adverse reactions. Another therapeutic option is required.

It can be challenging to distinguish FD symptoms from those of biliary dyspepsia, caused by biliary tract disease. Biliary dyspepsia is defined as biliary colic of no known organic, systemic, or metabolic origin. According to the Rome IV criteria, biliary colic is associated with steady pain in the right upper quadrant and/or epigastric area persisting for at least 30 min [5]. As epigastric pain can develop in patients with either biliary dyspepsia or FD, and as standard blood and imaging test results are normal in both, it is difficult to distinguish the disorders only using such tests [5,6]. Therefore, although epigastric or upper abdominal discomfort is a Rome IV criterion for FD, it is difficult to eliminate the possibility that the symptoms are at least partly attributable to biliary dyspepsia [7].

The CNU^®^ ((CNU); Myungmoon Pharm. Co., Seoul, Korea) capsule is a litholytic agent used to treat gallstones—namely, the trihydrated magnesium salts of chenodeoxycholic and ursodeoxycholic acid [8,9]. CNU improves the symptoms and gallbladder ejection fraction (GBEF) of patients with biliary dyspepsia [7,10]. Thus, we conducted a clinical trial that examined the efficacy and safety of CNU in patients with RFD.

## 2. Patients and Methods

### 2.1. Patients

This was a nonrandomized, multicenter, prospective, single-sided preliminary clinical trial. Six tertiary medical centers participated. The protocol conformed to the ethical guidelines of the World Medical Association Declaration of Helsinki and was approved by the institutional review boards of all participating facilities (general approval no. 320180300) and registered at ClinicalTrials.gov (NCT03844100) and cris.nih.go.kr (KCT0003608), accessed on 1 November 2019. All patients were aged ≥19 years and had FD diagnosed according to the Rome IV criteria (including symptoms for at least 3 months and symptom onset at least 6 months prior to diagnosis) [11]. All patients also met the RFD criteria (continuous symptoms disrupting daily life (7-point overall symptom scale score ≥5) for at least 6 months and a failure to respond to at least two medical treatments) [6]. The exclusion criteria were any organic disease of the pancreaticobiliary system, liver, kidneys, or gastrointestinal (GI) tract revealed by laboratory tests, ultrasonography, or radiography; diabetes, thyroid disease, connective tissue disease, a mental condition, or another systemic disease; a disorder of the central nervous system (cerebral hemorrhage or infarction with a residual deficit) or the autonomic nervous system; previous abdominal surgery; endoscopy-confirmed or a history of a GI ulcer, Helicobacter pylori eradication, erosion, tumor, another organic disease, or esophagitis; reduced gallbladder contractility (GBEF <40%); pregnancy or lactation; and/or the use of a medication increasing biliary cholesterol secretion or the potential to induce a hepatotoxic effect. All enrolled patients underwent DISIDA scans to measure the GBEF values [12]. Serial hepatobiliary analog images were obtained at 5, 10, 20, 30, 45, and 60 min after the intravenously injection of 8 mCi DISIDA and 30 min after the ingestion of milk. The GBEF was derived by calculating the counts in the GB before and 30 min after the ingestion of milk.

### 2.2. Medication and Follow-Up

All study participants added one 250 mg CNU capsule orally twice daily (in the morning and evening, either with or after a meal) to their existing FD medication for 12 weeks. All returned to the hospital 4 and 12 weeks after medication commencement for monitoring of the adherence to therapy and complications and for laboratory tests. The drug adherence means the proportion between the number of regular assumption days and the total days. If a medication-related complication was suspected, the medication was immediately stopped, and a full evaluation was conducted before resuming the medication. The 7-point global symptom scale score was measured before and after the 12-week course to assess symptom improvement [13].

### 2.3. Study Endpoints

The primary study endpoint was to determine the drug efficacy, evaluating the proportion of patients evidencing symptom improvement after completing the 12-week course. The secondary endpoints focused on safety assessed by measurement of the vital signs, laboratory tests, and monitoring of abnormal clinical responses.

### 2.4. Statistical Analysis

Continuous variables were reported as means ± standard deviations, and categorical variables were given as numbers (percentages). The mean pre- and posttreatment global symptom scale scores were compared using the Student’s *t*-test. All statistical analyses were performed using SPSS ver. 23.0 (IBM Corp., Armonk, NY, USA). A two-tailed *p*-value <0.05 was considered statistically significant.

## 3. Results

A total of 52 patients were prospectively screened at six institutions (Figure 1). Twelve were excluded because of low gallbladder contractility (*n* = 5), refusal to provide consent (*n* = 5), concomitant use of prohibited drugs (*n* = 1), and the presence of cholelithiasis (*n* = 1). The final analysis was performed with 37 patients after excluding 3 patients who subsequently withdrew consent.

The mean age was 51.3 years (14 males and 23 females; Table 1). The mean height was 163.5 cm, mean weight was 61.8 kg, and mean body mass index was 22.9 kg/m^2^. Eight (21.6%) were postmenopausal. The mean systolic blood pressure was 120.3 mm Hg, mean diastolic blood pressure was 75.5 mm Hg, and mean heart rate was 76.3 beats/min. The mean number of symptoms prior to CNU administration was 2.4. The symptoms were epigastric pain (*n* = 24, 64.9%), epigastric burning (*n* = 15, 40.5%), postprandial fullness (*n* = 30, 81.1%), and early satiety (*n* = 21, 56.8%). The mean symptom duration was 48.2 months.

Before CNU commencement, the mean number of FD drugs taken was 3.3 (Table 2). A total of 11 types of drugs were used; the most common were prokinetics (81.1%).

No patient exhibited an abnormal vital sign or blood test result before commencing CNU (Table 3). The mean GBEF in a di-isopropyl iminodiacetic acid (DISIDA) scan before CNU administration was 64.8%, reflecting the exclusion of patients with GBEFs <40%. No GB stone or GB sludge were observed during ultrasonography before enrolling in this study.

The treatment completion rate was 92.5%. The mean duration of CNU medication was 12 weeks. Adherence to the test drug was 80–89% in 10 patients and 90–100% in 27; the mean adherence was 95.3%. The mean global symptom scale score significantly decreased from 5.6 before CNU commencement to 2.6 at the end of the 12-week study period (*p* = 0.001) (Figure 2). The symptoms improved in 94.6% of patients and remained the same in 5.4%. No patients evidenced worsened or unevaluable symptoms (Table 4).

No patients exhibited an abnormal vital sign or blood test result at 4 or 12 weeks after the initiation of therapy. A total of 11 adverse events occurred (30%; Table 5), of which 8 were of grade 1 and 3 were of grade 2. Only one event (grade 1 diarrhea) was (possibly) associated with the drug. The diarrhea resolved when CNU was temporarily stopped; after which, CNU resumed. Other adverse events were vaginal inflammation, Tinea inguinalis, and ligament rupture, which were managed by non-GI medication. Asymptomatic hepatic hemangioma was incidentally also found. The remaining events were unlikely to be, or clearly not, related to the study drug.

## 4. Discussion

RFD is characterized by FD symptoms that persist for at least 6 months and do not respond to at least two medical treatments (acid suppressants, proton pump inhibitors, prokinetic agents, or *Helicobacter pylori* eradication therapy) [6]. RFD accounts for approximately 24% of all FD [14]. Psychological treatment is effective in the short term but not in the long term; the cost-effectiveness of such therapy is doubtful [6]. Research into causes other than GI etiologies of RFD is required, as are alternative treatments.

We found that CNU improved the symptoms of 94.6% of RFD patients who had not responded to prior FD treatments. This suggests that a substantial proportion of patients with RFD likely have biliary, not gastroduodenal, dyspepsia, and patients of biliary origin were intensively recruited in this study, resulting in a relatively high response rate. The frequency of functional (acalculous) biliary-type pain is as high as 7.6% in men and 20.7% in women [15]. Gallbladder dysfunction may reflect a generalized dysmotility disorder, such as irritable bowel syndrome, chronic constipation, or (perhaps) gastroparesis [16,17]. In a previous study, 22.2% of RFD patients evidenced biliary dyspepsia and gallbladder dyskinesia [7]. Postprandial pain in the epigastrium, bloating, dyspepsia, and nausea can develop in patients with either FD or gallbladder dyskinesia [18]. Although the latter is rather rare, the principal presenting symptoms (pain in the upper right abdomen and epigastrium) are not easy to distinguish from those of very common conditions such as gastroesophageal reflux disease, irritable bowel syndrome, and FD [19]. As the symptoms of gallbladder dyskinesia can be mistaken for those of FD, the evaluation of gallbladder function should be considered in patients with RFD [7].

Biliary-type pain is thought to be caused principally by gallbladder dyskinesia and dysfunction of the sphincter of Oddi, because the pain is often successfully treated via cholecystectomy or sphincterotomy (sphincter ablation) in cases lacking recognized organic causes [5]. However, the symptoms of some patients do not improve even after cholecystectomy and/or sphincterotomy, suggesting that gallbladder dyskinesia and sphincter of Oddi dysfunction cannot explain all biliary-type pain. In addition, such patients may complain of discomfort rather than pain and biliary dyspepsia. Biliary pain (i.e., biliary colic) is defined (by the Rome IV criteria) as pain persisting for longer than 30 min. Some patients report recurrent discomfort that does not meet this criterion.

Although the pathogenesis of functional biliary-type pain and the mechanism of biliary dyspepsia thus remains unclear, various possibilities have been considered. First, a narrowed cystic duct can render gallbladder emptying incomplete, eventually triggering chronic cholecystitis and biliary pain [20]. Ruffolo et al. reported an association between the cystic duct (rather than the common bile duct or sphincter of Oddi) and gallbladder dysfunction [21]. In addition, microlithiasis is associated with gallbladder dyskinesia, and the GBEF has been reported to be significantly lower in patients with microlithiasis compared to the controls [22]. Beyond these potential mechanisms, several factors, such as prostaglandin E2, are also associated with the pathogenesis of gallbladder dyskinesia [5,23,24]. Of these, ursodeoxycholic acid may enhance microlithiasis dissolution and, thus, improve the dyskinesia [10]. However, further analysis of the bile from patients treated with CNU is warranted to evaluate the effects of such treatment on microlithiasis.

Functional gallbladder disorder (FGBD) is characterized by biliary pain in the absence of gallbladder stones, sludge, or any other structural pathology [5]. Supportive findings include a low GBEF in gallbladder scintigraphy and normal levels of liver enzymes, conjugated bilirubin, and amylase/lipase. However, a low GBEF is not required for a diagnosis of FGBD; this is not a specific finding [25]. In a previous study, the rate of biliary dyspepsia caused by gallbladder dyskinesia was 22.2%, and the symptoms improved when the dyskinesia improved [7]. In the current study, we included only patients with normal GBEFs (≥40% on the DISIDA scan); we excluded patients with biliary dyspepsia caused by gallbladder dyskinesia. The causes and pathological mechanisms of biliary pain or biliary dyspepsia in patients with normal gallbladder contractability have not been well-studied.

Biliary dyspepsia can be viewed in two ways. Gallbladder dysmotility may play a role in its pathogenesis by promoting gallbladder inflammation. Intra-gallbladder microlithiasis can trigger a vicious cycle between the bile stasis and inflammation [5]. Gallbladder microlithiasis is associated with a decreased GBEF [22]. The abnormalities of patients with microlithiasis can be remedied via litholytic therapy. Symptom improvement after a prescription of litholytic agents in patients with biliary dyspepsia is thought to indicate reduced biliary microlithiasis. Such treatment would also be expected to increase the GBEF, reducing both the bile stasis and gallbladder inflammation [7,9,10].

We found that litholytic agents improved the symptoms of biliary dyspepsia. Routine upper endoscopy, medical imaging, and laboratory tests fail to demonstrate abnormalities in patients with either FD or gallbladder dyskinesia [5,11]. A DISIDA scan is required to detect gallbladder dyskinesia in RFD patients. In such patients, litholytic and choleretic agents may relieve symptoms by improving the GBEF [7,10]. Even in RFD patients with normal GBEFs, litholytic agents may relieve biliary dyspepsia caused by microlithiasis.

Our work had certain limitations. First, this was not a randomized trial, and subjects were not consecutively enrolled. The sample size was rather small. Recruitment of a high number of patients was difficult, because we imposed strict inclusion and exclusion criteria. In this study, the effect of a placebo could not be confirmed, because comparison with a placebo could not be performed. Further randomized controlled trials, including placebos, are needed in the future. Second, sphincter of Oddi dysfunction could not be excluded, because the sphincter pressure was not measured. Such dysfunction is rare in the East; manometry is both invasive and nonstandard, and the predictive utility of a pressure datum in terms of dysfunction is controversial [5]. However, sphincter of Oddi dysfunction was unlikely; bile duct dilation was not apparent on the abdominal imaging, and the litholytic treatment was very successful. Finally, microlithiasis (which is thought to cause biliary dyspepsia) was not directly measured; the precise cause of dyspepsia remains unidentified. Although endoscopic retrograde cholangiopancreatography would have confirmed the biliary tract microlithiasis, we did not perform this test, because it is invasive and associated with potential complications.

## 5. Conclusions

RFD patients who do not respond to conventional medications include individuals with FD of biliary, rather than GI, origin. In such patients, the symptoms improve when a litholytic agent reduces the biliary microlithiasis. A large-scale clinical study is required, as is additional research on the mechanism of RFD and the effectiveness of litholytic agents.

## Figures and Tables

**Figure 1 jcm-11-03190-f001:**
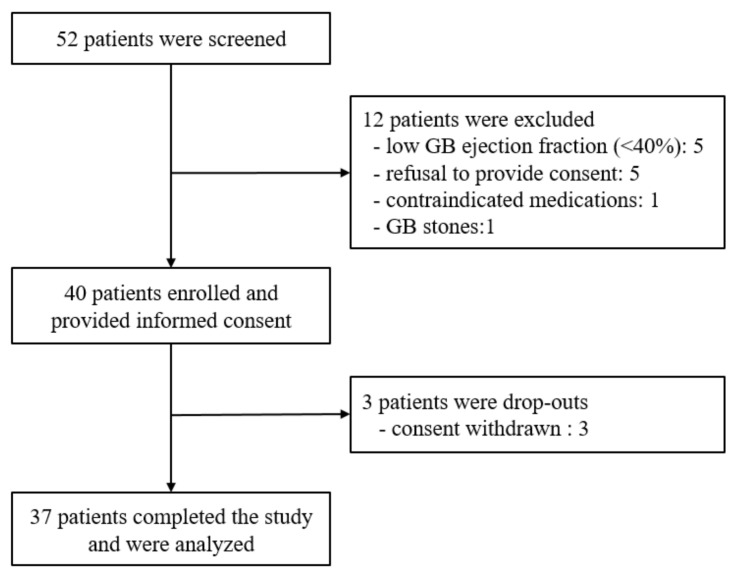
Patient flowchart. GB, gallbladder.

**Figure 2 jcm-11-03190-f002:**
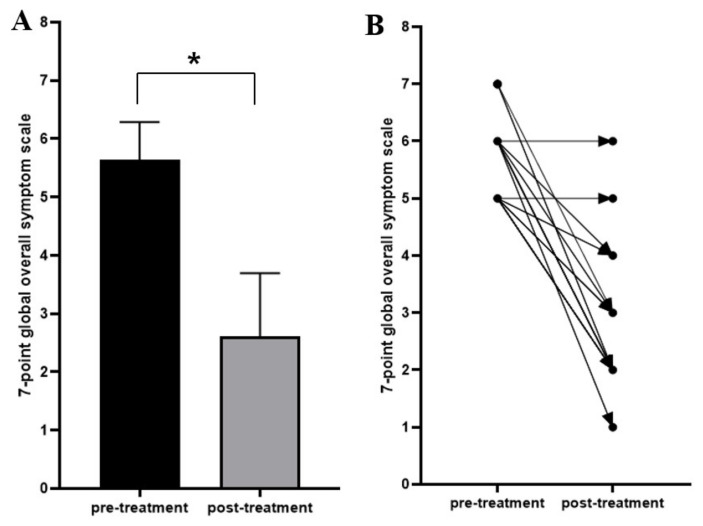
The 7-point global overall symptom scale scores before and after litholytic agent treatment (*n* = 37). (**A**). The mean 7-point global overall symptom scale scores decreased significantly posttreatment compared with pretreatment. (**B**). Paired dot plot of the pre- and posttreatment 7-point global overall symptom scale scores for each patient, showing improved scores in 35 patients and unchanged scores in two patients. * *p* = 0.001.

**Table 1 jcm-11-03190-t001:** Basic patient characteristics.

Variables	Values (*n* = 37)
**Basic Characteristics**	
Age (y), mean ± SD	51.3 ± 14.6
Sex (male:female), *n* (%)	14:23
Height (m), mean ± SD	163.5 ± 9.6
Weight (kg), mean ± SD	61.8 ± 12.0
BMI (kg/m^2^), mean ± SD	22.9 ± 2.8
Menopause, *n* (%)	8 (21.6)
Systolic BP (mm Hg), mean ± SD	120.3 ± 11.9
Diastolic BP (mm Hg), mean ± SD	75.5 ± 9.3
Heart rate (beats/min), mean ± SD	76.3 ± 10.2
**Symptoms**	
**Number of Symptoms, Mean ± SD**	2.4 ± 0.9
**Type of Symptoms, *n* (%)**	
Epigastric pain	24 (64.9)
Epigastric burning	15 (40.5)
Postprandial fullness	30 (81.1)
Early satiety	21 (56.8)
**Duration (Months), Mean ± SD (Range)**	48.2 ± 57.8 (7–240)

BMI, body mass index; BP, blood pressure; SD, standard deviation.

**Table 2 jcm-11-03190-t002:** Medications for dyspepsia prior to litholytic agent administration.

Medication Number and Type	Values (*n* = 37)
**Number of Medications, Mean ± SD**	3.3 ± 1.3
**Type of Medication, *n* (%)**	
Prokinetics	30 (81.1)
Digestive enzymes	20 (54.1)
Proton pump inhibitors	16 (43.2)
H2-receptor antagonists	15 (40.5)
Gastric mucosa protective agents	13 (35.1)
Antacids	10 (27.0)
Anticholinergics	8 (21.6)
Antidepressant agents	3 (8.1)
Probiotics	3 (8.1)
Analgesics	2 (5.4)
Antibiotics	1 (2.7)

H2, histamine 2; SD, standard deviation.

**Table 3 jcm-11-03190-t003:** Laboratory findings and gallbladder ejection fraction prior to litholytic agent administration.

Variables	Values (*n* = 37)	Reference Ranges
**Laboratory Findings, Mean ± SD**		
T3 (ng/dL)	81.2 ± 39.7	71–161
T4 (ng/dL)	7.0 ± 2.4	5.5–10.6
TSH (µIU/mL)	2.1 ± 1.3	0.86–4.6
HbA1c (%)	5.5 ± 0.3	4.8–6.3
White blood cell count (10^3^/µL)	6.4 ± 1.6	4.0–10.8
Neutrophil (%)	54.1 ± 14.5	40–73
Red blood cell count (10^6^/µL)	4.4 ± 0.4	4.0–5.4
Hemoglobin (g/dL)	13.6 ± 1.0	13–17
SGOT (IU/L)	21.3 ± 5.0	16–37
SGPT (IU/L)	18.9 ± 10.4	11–46
Total bilirubin (mg/dL)	0.7 ± 0.4	0.3–1.3
Direct bilirubin (mg/dL)	0.2 ± 0.1	0.1–0.3
γ-GTP (IU/L)	19.4 ± 9.3	8–46
Alkaline phosphatase (IU/L)	64.2 ± 15.4	44–99
Sodium (mmol/L)	139.8 ± 2.7	138–146
Potassium (mmol/L)	4.3 ± 0.4	3.6–4.8
**GBEF (%), Mean ± SD**	64.8 ± 13.4	-

γ–GTP, γ-glutamyl transpeptidase; GBEF, gallbladder ejection fraction; HbA1c, glycosylated hemoglobin; SD, standard deviation; SGOT, serum glutamic-oxaloacetic transaminase; SGPT, serum glutamic-pyruvate transaminase; T3, triiodothyronine; T4, thyroxine; TSH, thyroid stimulating hormone.

**Table 4 jcm-11-03190-t004:** General symptom changes after treatment with a litholytic agent (*n* = 37).

Symptom Change	Patients, *n* (%)
Improved	35 (94.6)
Unchanged	2 (5.4)
Worsened	0
Unevaluable	0

**Table 5 jcm-11-03190-t005:** Adverse events.

Patient No.	Event	Grade †	Management	Result	Relationship with CNU ‡
1	Abdominal pain	2	None	Symptom disappeared	Not related
2	Diarrhea	1	Stop medication	Symptom disappeared	Possibly related
3	Dyspepsia	2	Add other GI medications	Symptom disappeared	Not related
4	Non-cardiac chest pain	1	Add other GI medications	Symptom disappeared	Unlikely related
5	Vaginal inflammation	1	Add other non-GI medications	Symptom continued	Not related
6	Pain (epigastric)	1	None	Symptom disappeared	Not related
7	Tinea inguinalis	2	Add other non-GI medications	Symptom disappeared	Not related
8	Diarrhea	1	None	Symptom disappeared	Unlikely related
9	Ligament rupture (right ankle)	1	Add other non-GI medications	Symptom disappeared	Not related
10	Diarrhea	1	None	Symptom disappeared	Unlikely related
11	Hepatic hemangioma	1	None	Symptom disappeared	Not related

† Classified according to the Common Terminology Criteria for Adverse Events 4.03 grade. ‡ Relationships between complications and medications were assessed according to the Naranjo algorithm (1981). GI, gastrointestinal.

## Data Availability

The data that support. the findings of this study are available from the corresponding author upon reasonable request.

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
