# Peer review of "Efficacy of Chenodeoxycholic Acid and Ursodeoxycholic Acid Treatments for Refractory Functional Dyspepsia"

_jcm, 2022, doi:10.3390/jcm11113190_

Round 1
Reviewer 1 Report
Thank you for the privilege of reviewing your work. This manuscript is well written. While interesting, the manuscript needs some improvement.
1. Please describe the accumulation period.
2. Did you set up the statistics before starting the study?
3. What is the treatment completion rate?
4. Can you say that you are not involved in each of these events in adverse events?
(for example, dyspepsia.)
5. The authors must fill in limitation that the placebo effect resulted in a favorable outcome.
6. I think the 95% response rate is too high. You need to write a little more rationale
Author Response
▣ For Reviewer 1
We appreciate your thoughtful comments regarding our manuscript.
▣ Evaluations
Reviewer Comments:
Reviewer 1
Thank you for the privilege of reviewing your work. This manuscript is well written. While interesting, the manuscript needs some improvement.
- Please describe the accumulation period.
Answer) The mean duration of clinical symptoms before CNU medication was 48.2 months. And according to the protocol of this study, the patient's CNU medication was 12 weeks. The content of this is expressed in the text as follows.
In Result section
“The mean symptom duration was 48.2 months.”
“The mean duration of CNU medication was 12 weeks.”
- Did you set up the statistics before starting the study?
Answer) This prospective study is a one-arm exploratory study, conducted with the concept of a pilot study, and there is no statistical analysis protocol for the number of patients.
- What is the treatment completion rate?
Answer) 40 people were enrolled and took medication, and 37 people completed the treatment. Therefore, the treatment completion rate is 92.5%. We have added these to the text as follows:
In Result section
“The treatment completion rate is 92.5%.”
- Can you say that you are not involved in each of these events in adverse events?
(for example, dyspepsia.)
Answer) This study was a multicenter study conducted at 6 institutions, and each adverse event was reported to the IRB of each institution. Therefore, there was no room for involvement in adverse events.
- The authors must fill in limitation that the placebo effect resulted in a favorable outcome.
Answer) As this study was a one-arm study that did not compare with placebo, as pointed out by the reviewer, the limitations were added as follows.
“In this study, the effect of placebo could not be confirmed because comparison with placebo could not be performed. Further randomized controlled trial including placebo is needed in the future.”
- I think the 95% response rate is too high. You need to write a little more rationale
Answer) This study was conducted on patients whose functional dyspepsia did not resolve with existing drugs, so it is highly likely that patients with biliary origin were intensively recruited. It is thought that the response rate was high because these patients were targeted. However, since this study was conducted on small-sized patients, it is necessary to verify the response rate by conducting large-scale patients in the future.
The content of this is expressed in the text as follows.
“and patients with biliary origin were intensively recruited in this study resulting relatively high response rate”

Reviewer 2 Report
The paper "Efficacy of Chenodeoxycholic Acid and Ursodeoxycholic Acid Treatments for Refractory Functional Dyspepsia" is original and interesting. Functional dyspepsia is indeed difficult to treat in some patients, especially biliary dyspepsia. In regard of this, the paper helps the clinicians to improve the therapy.
However, it is not clear if the patients have microlithiazis. The GB stones are mentioned at the exclusion criteria. The ultrasound results should be mentioned.
Author Response
▣ For Reviewer 2
We appreciate your thoughtful comments regarding our manuscript.
▣ Evaluations
Reviewer Comments:
Reviewer 2
The paper "Efficacy of Chenodeoxycholic Acid and Ursodeoxycholic Acid Treatments for Refractory Functional Dyspepsia" is original and interesting. Functional dyspepsia is indeed difficult to treat in some patients, especially biliary dyspepsia. In regard of this, the paper helps the clinicians to improve the therapy.
However, it is not clear if the patients have microlithiazis. The GB stones are mentioned at the exclusion criteria. The ultrasound results should be mentioned.
Answer) The exclusion criteria of this study are specified as follows.
“The exclusion criteria were any organic disease of the pancreaticobiliary system, liver, kidneys, or gastrointestinal (GI) tract revealed by laboratory tests, ultrasonography, or radiography”
Patients enrolled in this study underwent a ultrasonography screening test and showed no GB stones or sludge.
As the reviewer pointed out, to clarify the meaning, the ultrasonography results are specified in the results as follows.
In Result section
“No GB stone or GB sludge was observed in the ultrasonography before enrolling in this study.”

Reviewer 3 Report
To the Authors:
this paper summarizes efficacy and safety of a therapy for a very frequently encountered problem. Studies describing the role of biliary acids in functional dyspepsia date back to the 80s and 90s but it is interesting re- underline they could be a potential therapeutic target. I ask You to revise some minor aspects.
Regarding the abstract:
please briefly specificy causes for biliary dispepsia.
Regarding the section "patients and methods":
I ask to the Authors to specify if obesity could have a pathogenetic role in biliary dispepsia and to specify if obese patients were excluded from the study.
Please also specify if Helicobacter Pylori eradication was checked and the method used (i.e. stool antigen? EGDS? other?)
In line 84 what does radiography mean (X- Ray test, Magnetic resonance imaging)? Did you evaluate presence of microlithiasis using cholangio-MR?
Pleas also biefly describe what the GBEF to calculate gallbladder contractility consists of.
How did You calculate drug aderence (i.e. proportion between the number of regular assumption days and the total days? other?)? Please report this aspect.
Regarding section "discussion":
Biliary dispepsia is a problem already faced in past where a quite high response to placebo was also reported Please specify this aspect. (i.e. https://pubmed.ncbi.nlm.nih.gov/8842844/ ). Are You interested in performing a randomized controlled- trial?
Biliary acids also play a role in modifying intestinal bacteria and mucosal integrity (i.e. https://pubmed.ncbi.nlm.nih.gov/31916349/ and https://pubmed.ncbi.nlm.nih.gov/32422942/). Did You evaluate this aspect? Could it be an addictional effect for symptoms control?
Please correct some typos or minor English Language mistakes as the following ones:
line 96: "...for monitoring adherence to therapy"
line 98: "...before resuming..."
line 102: "...was to determine drug effecicay, evaluating the proportion"
line 115: does prohibited stand for incompatible?
line 145: "...significantly decreased"
line 173: please specify "other conventional"
line 207: erase +
Regarding tables:
In Table 5 You describe three total cases of diarrhea. Please specify in the text why the other two cases are not associated with drug (i.e. infectious? Other resolved causes?). Moreover, regarding hepatic hemangioma You describe "symptom disappeared": what do You mean? It was asympromatic or it disappeared during radiologic follow- up?
Author Response
▣ For Reviewer 3
We appreciate your thoughtful comments regarding our manuscript.
▣ Evaluations
Reviewer Comments:
Reviewer 3
To the Authors:
this paper summarizes efficacy and safety of a therapy for a very frequently encountered problem. Studies describing the role of biliary acids in functional dyspepsia date back to the 80s and 90s but it is interesting re- underline they could be a potential therapeutic target. I ask You to revise some minor aspects.
Regarding the abstract:
- please briefly specificy causes for biliary dispepsia.
Answer) As the reviewer pointed out, the cause has been added as follows.
“The implicated causes of functional biliary dyspepsia are narrowed cystic duct, Sphincter of Oddi dysfuction , microlithiasis and gallbladder dyskinesia.”
Regarding the section "patients and methods":
- I ask to the Authors to specify if obesity could have a pathogenetic role in biliary dispepsia and to specify if obese patients were excluded from the study.
Answer) The implicated causes of functional biliary dyspepsia are narrowed cystic duct, Sphincter of Oddi dysfuction, microlithiasis and gallbladder dyskinesia.
Obesity was not presumed as a pathogenetic mechanism in biliary dyspepsia, so it was not excluded from this study.
- Please also specify if Helicobacter Pylori eradication was checked and the method used (i.e. stool antigen? EGDS? other?)
Answer) The following were added to the exclusion criteria of this study.
The exclusion criteria were any organic disease of the pancreaticobiliary system, liver, kidneys, or gastrointestinal (GI) tract revealed by laboratory tests, ultrasonography, or radiography; diabetes, thyroid disease, connective tissue disease, a mental condition, or another systemic disease; a disorder of the central nervous system (cerebral hemorrhage or infarction with a residual deficit) or the autonomic nervous system; previous abdominal surgery; endoscopy-confirmed or a history of a GI ulcer, Helicobacter pylori eradication, erosion, tumor, another organic disease, or esophagitis; reduced gallbladder contractility (GBEF < 40%); pregnancy or lactation; and/or use of a medication increasing biliary cholesterol secretion or the potential to induce a hepatotoxic effect.
- In line 84 what does radiography mean (X- Ray test, Magnetic resonance imaging)? Did you evaluate presence of microlithiasis using cholangio-MR?
Answer) Not all imaging tests were performed for each disease. By referring to the types of imaging tests, ultrasonography was performed to differentiate GB stones.
The exclusion criteria of this study are specified as follows.
“The exclusion criteria were any organic disease of the pancreaticobiliary system, liver, kidneys, or gastrointestinal (GI) tract revealed by laboratory tests, ultrasonography, or radiography”
Patients enrolled in this study underwent a US screening test and showed no GB stones or sludge.
As the reviewer pointed out, to clarify the meaning, the US results are specified in the results as follows.
In Result section
“No GB stone or GB sludge was observed in the ultrasonography before enrolling in this study.”
Microlithiasis can be observed microscopically, and this study could not confirm it in the patient's bile, which may be a limitation of this study.
The limitation on this is mentioned below.
“Finally, microlithiasis (which is thought to cause biliary dyspepsia) was not directly measured; the precise cause of dyspepsia remains unidentified. Although endoscopic retrograde cholangiopancreatography would have confirmed biliary tract microlithiasis, we did not perform this test because it is invasive and associated with potential complications.”
- Pleas also biefly describe what the GBEF to calculate gallbladder contractility consists of.
Answer) All enrolled patients underwent DISIDA scans to measure GBEF values.1 Each patient was given 8 mCi DISIDA intravenously under a large-field-of-view gamma camera. Serial hepatobiliary analogue images were obtained at 5, 10, 20, 30, 45, and 60 minutes after the injection or until the GB was adequately filled. The patients immediately drank 200 mL milk containing about 13 g fat after completion of the filling phase. Analogue images were recorded 30 minutes after ingestion of the milk. The GBEF was derived by calculating the counts in the GB before and 30 minutes after ingestion of the milk.
Reference 1. DiBaise JK, Richmond BK, Ziessman HA, et al. Cholecystokinin-cholescintigraphy in adults: consensus recommendations of an interdisciplinary panel. Clin Nucl Med 2012;37:63-70.
I have added these contents as follows.
“All enrolled patients underwent DISIDA scans to measure GBEF values.1 Serial hepatobiliary analogue images were obtained at 5, 10, 20, 30, 45, and 60 minutes after the intravenously injection of 8 mCi DISIDA and 30 minutes after ingestion of the milk. The GBEF was derived by calculating the counts in the GB before and 30 minutes after ingestion of the milk.”
- How did You calculate drug aderence (i.e. proportion between the number of regular assumption days and the total days? other?)? Please report this aspect.
Answer) The drug adherence means proportion between the number of regular assumption days and the total days.
I have added these contents as follows.
“The drug adherence means proportion between the number of regular assumption days and the total days.”
Regarding section "discussion":
- Biliary dispepsia is a problem already faced in past where a quite high response to placebo was also reported Please specify this aspect. (i.e. https://pubmed.ncbi.nlm.nih.gov/8842844/ ). Are You interested in performing a randomized controlled- trial?
Answer) As this study was a one-arm study that did not compare with placebo, as pointed out by the reviewer, the limitations were added as follows. In addition, this research team intends to conduct a randomized controlled-trial.
“In this study, the effect of placebo could not be confirmed because comparison with placebo could not be performed. Further randomized controlled trial including placebo is needed in the future.”
- Biliary acids also play a role in modifying intestinal bacteria and mucosal integrity (i.e. https://pubmed.ncbi.nlm.nih.gov/31916349/ and https://pubmed.ncbi.nlm.nih.gov/32422942/). Did You evaluate this aspect? Could it be an addictional effect for symptoms control?
Answer) We could not confirm these contents in this study, but it is considered to be a good research topic. It would be good to conduct further research on this.
- Please correct some typos or minor English Language mistakes as the following ones:
line 96: "...for monitoring adherence to therapy"
line 98: "...before resuming..."
line 102: "...was to determine drug effecicay, evaluating the proportion"
line 115: does prohibited stand for incompatible?
line 145: "...significantly decreased"
line 173: please specify "other conventional"
line 207: erase +
Answer) I have edited the text as you pointed out.
Regarding tables:
- In Table 5 You describe three total cases of diarrhea. Please specify in the text why the other two cases are not associated with drug (i.e. infectious? Other resolved causes?). Moreover, regarding hepatic hemangioma You describe "symptom disappeared": what do You mean? It was asympromatic or it disappeared during radiologic follow- up?
Answer) Table 5 presents a collection of all adverse events reported to the IRB that occurred in patients enrolled in this study. The infection occurred in vigina and improved after use of antibiotics. Tinea inguinalis improved after dermatological medication. Hepatic hemangioma was discovered during the study and was an incidental finding with no symptoms.
The text is expressed as follows.
“Other adverse events are vaginal inflammation, Tinea inguinalis, and ligament rupture, which were managed by non-GI medication. And asymptomatic hepatic hemangioma was found incidentally.”

Round 2
Reviewer 1 Report
Thank you for modifying the manuscript per suggestions. The manuscript has been revised well